# MDPrePost-Net: A Spatial-Spectral-Temporal Fully Convolutional Network for Mapping of Mangrove Degradation Affected by Hurricane Irma 2017 Using Sentinel-2 Data

Ilham Jamaluddin [1], Tipajin Thaipisutikul [2], Ying-Nong Chen [1,3,*], Chi-Hung Chuang [4] and Chih-Lin Hu [5]

1   Center for Space and Remote Sensing Research, National Central University, No. 300, Jhongda Rd., Jhongli Dist., Taoyuan City 32001, Taiwan; ilhamjamaluddin09@gmail.com
2   Faculty of Information and Communication Technology, Mahidol University, 999 Phuttamonthon 4 Rd., Salaya, Nakhon Pathom 73170, Thailand; tipajin.tha@mahidol.ac.th
3   Department of Computer Science and Information Engineering, National Central University, No. 300, Jhongda Rd., Jhongli Dist., Taoyuan City 32001, Taiwan
4   Department of Applied Informatics, Fo Guang University, No. 160, Linwei Rd., Jiaoxi Township, Yilan City 262307, Taiwan; chchuang@mail.fgu.edu.tw
5   Department of Communication Engineering, National Central University, No. 300, Jhongda Rd., Jhongli Dist., Taoyuan City 32001, Taiwan; chihlin.hu@gmail.com
*   Correspondence: yingnong1218@csrsr.ncu.edu.tw

**Abstract:** Mangroves are grown in intertidal zones along tropical and subtropical climate areas, which have many benefits for humans and ecosystems. The knowledge of mangrove conditions is essential to know the statuses of mangroves. Recently, satellite imagery has been widely used to generate mangrove and degradation mapping. Sentinel-2 is a volume of free satellite image data that has a temporal resolution of 5 days. When Hurricane Irma hit the southwest Florida coastal zone in 2017, it caused mangrove degradation. The relationship of satellite images between pre and post-hurricane events can provide a deeper understanding of the degraded mangrove areas that were affected by Hurricane Irma. This study proposed an MDPrePost-Net that considers images before and after hurricanes to classify non-mangrove, intact/healthy mangroves, and degraded mangroves classes affected by Hurricane Irma in southwest Florida using Sentinel-2 data. MDPrePost-Net is an end-to-end fully convolutional network (FCN) that consists of two main sub-models. The first sub-model is a pre-post deep feature extractor used to extract the spatial–spectral–temporal relationship between the pre, post, and mangrove conditions after the hurricane from the satellite images and the second sub-model is an FCN classifier as the classification part from extracted spatial–spectral–temporal deep features. Experimental results show that the accuracy and Intersection over Union (IoU) score by the proposed MDPrePost-Net for degraded mangrove are 98.25% and 96.82%, respectively. Based on the experimental results, MDPrePost-Net outperforms the state-of-the-art FCN models (e.g., U-Net, LinkNet, FPN, and FC-DenseNet) in terms of accuracy metrics. In addition, this study found that 26.64% (41,008.66 Ha) of the mangrove area was degraded due to Hurricane Irma along the southwest Florida coastal zone and the other 73.36% (112,924.70 Ha) mangrove area remained intact.

**Keywords:** fully convolutional network; convolutional long short-term memory (ConvLSTM); mangroves degradation; Hurricane Irma; Sentinel-2

## 1. Introduction

Mangroves are recognized as unique forms of vegetation in subtropical and tropical coastal zones in 118 countries and territories [1]. Mangrove vegetation provides many benefits for humans and surrounding ecosystems. For example, they provide carbon storage vegetation [2], coastal protection vegetation [3], breeding grounds [4], biodiversity conservation [4], and commercial vegetation [5]. Research revealed that 35% and 2.1% of

the global distribution of mangroves vanished during 1980–2000 [6] and 2000–2016 [7], respectively; also, the annual rate of mangrove loss was 0.26%–0.66% from 2000 to 2012 [8]. Because mangrove wetlands are among the few ecosystems with a large capacity for carbon storage, their mapping is essential.

The causes of mangrove loss can be both natural and anthropogenic. Human activities such as aquaculture, agriculture, hydrological pollution, port development, timber extraction, and urban development are some human causes of mangrove forest loss [9–11]. Extreme climate events [10], sea-level rise [12], and hurricanes [13] are some of the natural disasters that provoke mangrove forest loss. A principal factor behind biodiversity loss is ecosystem degradation [14]. Degradation can be defined as decreases in ecosystem composition, the provision of ecosystem services, productivity, and in vegetation canopy cover, which can all fundamentally alter an ecosystem [15–18]. Hurricanes are also among natural events that have substantially contributed to mangrove degradation [13,19]. In September 2017, southwest Florida was hit by one such storm, classed as a Category 3 hurricane, named Hurricane Irma [20]. Studies have reported on the Hurricane Irma–induced mangrove degradation in southwest Florida [20–22].

The mapping of mangrove degradation caused by Hurricane Irma is essential for understanding the status of mangroves at an affected area. The greatest challenge of mangrove mapping is to deal with the marred condition of mangrove forests and their wide distribution within a location. Remote sensing satellite imagery is advantageous because of a large observation scale and time-series data. Thus, it has been widely applied to mangrove mapping using traditional extraction methods or machine learning classification methods like pixel-based classification and object-based classification. The large scale of remote sensing data can be used to deal with the wide distribution of mangrove areas. By using remote sensing data, researchers were able to map mangroves without directly visiting all the mangrove areas. The remote sensing data also provide multispectral bands that can be used to get more information about the characteristics of mangrove objects. The combination of machine learning or deep learning with remote sensing data can increase effectiveness and reduce cost more efficiently than manual digitation techniques.

In recent years, several algorithms have been proposed for mangrove mapping. For example, Kamal et al. used an object-based algorithm for multiscale mangrove composition mapping and achieved overall accuracies around 82–94% [23]. Diniz et al. used the random forest (RF) algorithm for mangrove mapping in the Brazilian coastal zone over a period of three decades and achieved overall accuracies around 80–87% [24]. Moreover, Mondal et al. evaluated the accuracy of RF and classification and regression tree (CART) algorithms for mangrove mapping in South Africa and achieved an overall accuracy of RF and CART around 93.44% and 92.18%, respectively [25]. Chen also used the RF algorithm for mangrove mapping in Dongzhaigang, China, with Sentinel-2 imagery and achieved an overall accuracy of 90.47% [26]. Furthermore, some algorithms have been used for mangrove degradation mapping. For example, Lee et al. applied the RF algorithm as an ecological conceptual model for mapping mangrove degradation in Rakhine State, Myanmar (mangrove degradation mostly caused by anthropogenic disturbances) and Shark River in southwest Florida (mangrove degradation caused by Hurricane). The user's accuracy rates of the model in classifying degraded mangroves in Myanmar and Shark River were 69.3% and 73.9%, respectively [22]. McCarthy et al. applied decision tree (DT), neural network (NN), and support vector machine (SVM) classifiers for mapping mangrove degradation in areas affected by Hurricane Irma by using high-resolution satellite imagery. They reported that the user's accuracy rates for the DT classifier in classifying degraded mangroves were 62% and 56%, 75% for the NN classifier, and 59% for the SVM classifier [20,21]. Although the above studies can obtain sufficient user's accuracy rates on mangrove degradation mapping using specific algorithms, the accuracy can be improved with more appropriate techniques.

Recently, deep learning (DL) has been extensively applied to remote sensing satellite imagery [27–29]. As an important derivative of machine learning with multiple processing

layers, it can increase model performance and precision [30]. There are three sorts of DL algorithms for assessing satellite imagery, i.e., patch-based convolutional neural network (CNN), semantic segmentation or Fully Convolutional Network (FCN), and object detection algorithms [27,29]. Patch-based CNN and FCN algorithms have been used for mangrove classification using satellite imagery [31–33]. Hosseiny, et al. have proposed a spatio–temporal patch-based CNN, called WetNet, for wetland classification by using the LSTM layer in one of their architectures [34]. In 2014, a study introduced a fully convolutional network (FCN) that uses an encoder for feature extraction and a decoder to restore the input resolution by using deconvolutional or upsampling layers [35]. FCN has been widely used for pixel-based classification [29]. FCN uses an encoder for feature extraction and a decoder to restore the input resolution by deconvolutional or upsampling layers. A study reported that an FCN algorithm applied with Landsat imagery produced favorable results in mangrove mapping in large-scale areas [31]. An FCN algorithm is a pixel-based classification method, and such algorithms have been improved with advancements in CNN systems (e.g., AlexNet [36], VGG [37], ResNet [38], and DenseNet [39]) [29]. State-of-the-art FCN algorithms (e.g., U-Net [40], LinkNet [41], Me-Net [32], FC-DenseNet [42], and feature pyramid Net [43]) have recently drawn attention.

As above mentioned, a hurricane is one of the natural hazards that caused mangrove degradation in affected areas. This motivated us to exploit the relationship of satellite images between pre and post-hurricane events to better know the degraded mangrove area that was affected by Hurricane Irma. Thus, in this study, an end-to-end spatial-spectral-temporal FCN framework referred to as MDPrePost-Net is proposed to improve the classification accuracy of mangrove mapping. MDPrePost-Net consists of two main sub-models, a pre-post deep feature extractor, and an FCN classifier. Abundant previous studies have successfully applied convolutional LSTM (ConvLSTM) and FCN in remote sensing works, such as precipitation nowcasting [44], analyzing various types of input images for Side-Looking Airborne Radar (SLAR) imagery [45], and video frame sequences [46,47]. Therefore, in this study, the ConvLSTM is mainly used to extract spatial–spectral–temporal relationships from pre and post image data in the proposed MDPrePost-Net.

Sentinel-2 is a freely available passive remote sensing system that covers the entire planet and has 13 bands with different spatial resolutions (10, 20, and 60 m). Many studies have successfully applied machine learning and deep learning algorithms on Sentinel-2 for mangrove mapping [25,26,32]. Therefore, the proposed MDPrePost-Net uses Sentinel-2 imagery before and after the hurricane as the input data. Moreover, some spectral indices are also used in this study to improve model performance. In summary, this study has two aims: (1) to propose an end-to-end fully convolutional network that considers satellite imagery before and after hurricanes for mangrove degradation classification, referred to as MDPrePost-Net; (2) to use the MDPrePost-Net to draw a mangrove degradation map of the southwest Florida coastal zone affected by Hurricane Irma.

The rest of this paper is organized as follows: the materials and methods are discussed in Section 2. The results for mangrove degradation mapping from MDPrePost-Net in southwest Florida are discussed in Section 3. The discussion of our research is discussed in Section 4. Finally, in Section 5, the conclusions are drawn.

## 2. Materials and Methods

The work stages in this study are divided into data processing, classification processing, accuracy assessment, and mapping process, as shown in Figure 1. The data processing included collecting and correcting Sentinel-2 images from TOA to BOA surface reflectance, making spectral indices and combining with the original bands, making visually interpreted labels, and collecting reference samples for map accuracy assessments. The classification processing includes splitting the input data into training, validation, and testing data and classification training process. In the classification training processes, we proposed MDPrePost-Net and trained it using input data (Section 2.5). The accuracy assessment was divided into two parts, the first is to calculate algorithm output accuracy

by using the testing data (Section 2.6) and the second one is to calculate map accuracy by using collected reference samples (Section 2.7). The final stage is to extrapolate our proposed model to the whole study area and produce the final map of intact and degraded mangrove forests.

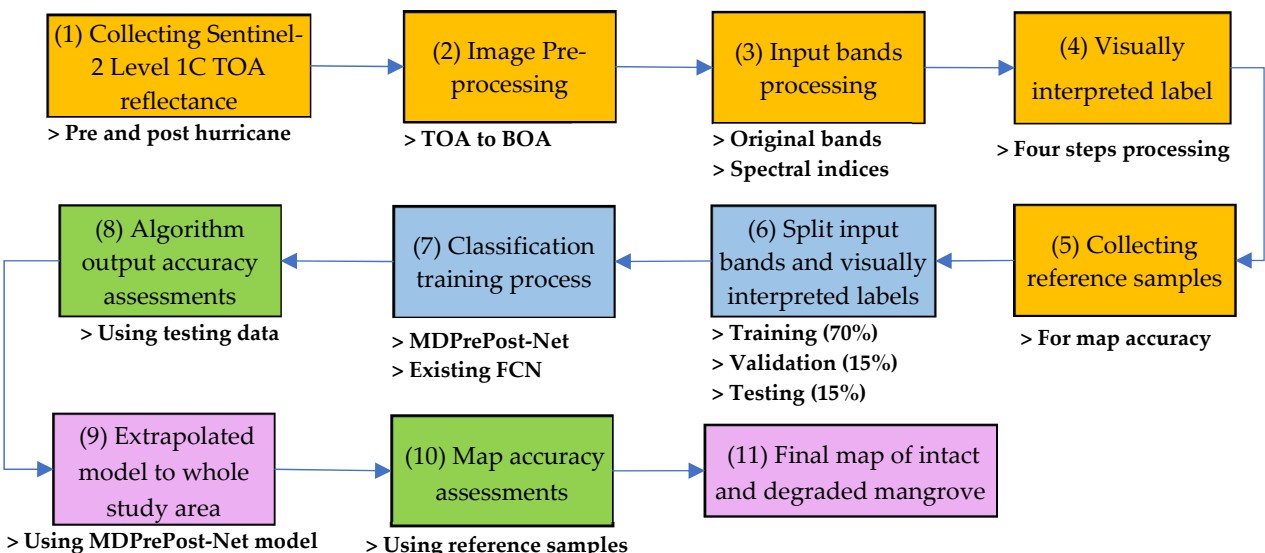

**Figure 1.** Flowchart of all work stages for data processing (orange box), classification process (blue box), accuracy assessments (green box), and map processing (purple box).

### 2.1. Study Area

We selected the coastal zones in southwest Florida, including the Everglades National Park and Rookery Bay National Estuarine Research Reserve (NERR), as shown in Figure 2. In this study area, there are only three mangrove species, namely *Rhizophora mangle*, *Avicennia germinans*, and *Laguncularia racemosa* [48], which are dominated by lagoonal, open coast, and estuarine mangroves [49]. Hurricane Irma in a Category-3 storm traversed this area in September 2017 and severely degraded the local mangrove ecosystem.

We selected seven areas of interest (AOIs) in the whole study area to formulate the input dataset of our model, as indicated by the purple and yellow boxes in Figure 1; the four AOIs in the purple box comprise intact/healthy mangroves, degraded mangroves, and non-mangrove areas, and those in the yellow box comprise non-mangrove areas (e.g., urban, waterbody, vegetation non-mangrove, and bare land). The total acreage of the study area was 1,211,312.53 Ha, and the total acreage of the 7 AOIs was 140,073.95 Ha.

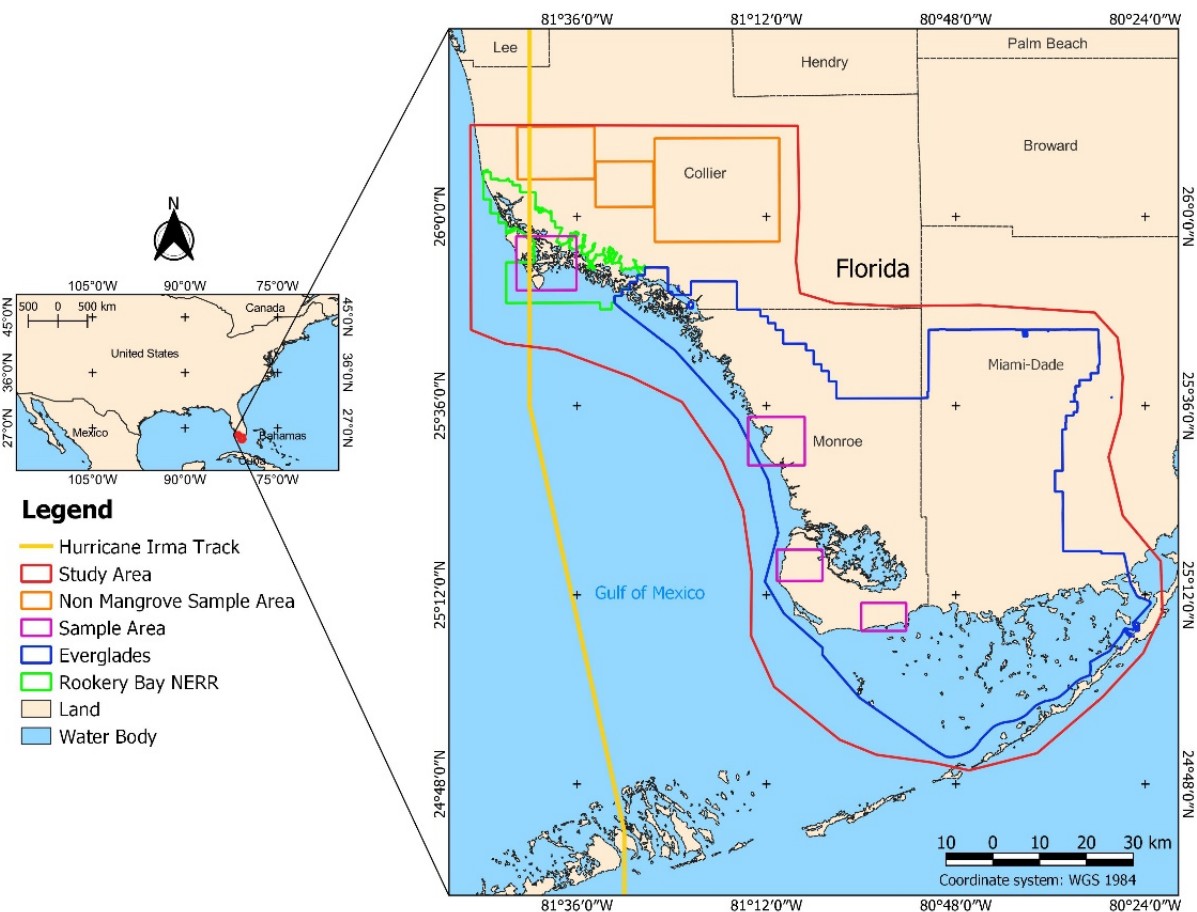

**Figure 2.** Whole study area in the southwest Florida coastal zone and Hurricane Irma track.

### 2.2. Sentinel-2 Pre-Processing

Sentinel-2 provides freely available satellite image data, courtesy of the European Space Agency (ESA). The characteristics of Sentinel-2 images are demonstrated in Table 1. Sentinel-2 has 13 bands: 4 bands with a 10-m spatial resolution, 6 with a 20-m spatial resolution, and 3 with a 60-m spatial resolution. We downloaded all Sentinel-2 images from the Sentinel Scientific Data Hub (https://scihub.copernicus.eu/dhus/, accessed on 1 December 2021). We selected Level-1C images (Top-of-Atmosphere (TOA) product with radiometric and geometric correction on a global reference system) with the lowest cloud cover in the downloading steps. Because Level-1C is a TOA product, we used sen2cor software (provided by the ESA) to correct the Level-1C images into Level-2A images (orthoimage Bottom-Of-Atmosphere (BOA) corrected reflectance product). After receiving the Sentinel-2 level-2A products, we resampled the SWIR-1 and SWIR-2 data to a 10-m spatial resolution to ensure a uniform spatial resolution in our entire dataset.

Regarding the input bands for the training model, we used the original bands from Sentinel-2 data as well as bands derived using spectral indices from previous research on mangrove mapping [24,25,31,32]. Spectral indices are generally mathematical formulas used to evaluate remote sensing image bands to enhance the spectral separability of the objects of interest. We calculated four spectral indices, namely the normalized difference vegetation index (NDVI) [50], combined mangrove recognition index (CMRI) [51], normalized difference mangrove index (NDMI) [52], and modular mangrove recognition index (MMRI) [24] (Table 2). These indices can contribute to the determination of mangrove and non-mangrove objects based on previous results. The NDVI is commonly used to evaluate various vegetation objects, including mangrove objects, in remote sensing [24,25,32]. The CMRI is derived by evaluating the differences between NDVI values and normalized dif-

ference water index (NDWI) values [53] to separate mangrove from non-mangrove objects and thus enhance the visibility of mangrove objects in the output result. The NDMI was designed to enhance the separability between mangrove and other vegetation types using the normalization difference between the Green and SWIR-2 bands that are associated with the characteristics of mangrove objects in satellite imagery. The MMRI was designed to enhance the contrast level between mangrove and non-mangrove objects and thus enable a swifter identification of mangrove objects. The MMRI is a combination of the NDVI and modified NDWI (MNDWI) [54]. A total of 10 input bands were used in this study (Blue, Green, Red, NIR, SWIR-1, SWIR-2, NDVI, CMRI, NDMI, and MMRI; Figure 3). The spectral indices were included in the band selection process to improve the classification performance, as discussed in Section 3.2.

**Table 1.** Characteristics of Sentinel-2 image.

| Band | Band Name | Central Wavelength (nm) | Spatial Resolution |
|------|-----------|-------------------------|--------------------|
| B1 | Aerosols | 442.3 | 60 |
| B2 | Blue | 492.1 | 10 |
| B3 | Green | 559 | 10 |
| B4 | Red | 665 | 10 |
| B5 | Red Edge 1 | 703.8 | 20 |
| B6 | Red Edge 2 | 739.1 | 20 |
| B7 | Red Edge 3 | 779.7 | 20 |
| B8 | Near Infrared (NIR) | 833 | 10 |
| B8A | Red Edge 4 | 864 | 20 |
| B9 | Water-vapor | 943.2 | 60 |
| B10 | Cirrus | 1376.9 | 60 |
| B11 | Shortwave Infrared (SWIR1) | 1610.4 | 20 |
| B12 | Shortwave Infrared (SWIR2) | 2185.7 | 20 |

**Table 2.** Spectral indices formula.

| Spectral Index | Formula | Reference |
|----------------|---------|-----------|
| NDVI | $(NIR - Red)/(Nir + Red)$ | [50] |
| NDWI | $(Green - NIR)/(Green + NIR)$ | [53] |
| CMRI | $(NDVI - NDWI)$ | [51] |
| NDMI | $(SWIR2 - Green)/(SWIR2 + Green)$ | [52] |
| MNDWI | $(Green - SWIR1)/(Green + SWIR2)$ | [54] |
| MMRI | $(|MNDWI| - |NDVI|)/(|MNDWI| + |NDVI|)$ | [24] |

### 2.3. Input Data for Model

Our proposed MDPrePost-Net is a type of supervised classification that uses the visually interpreted or labeled image data as the target data. The input data were divided into spectral input bands from Sentinel-2 and target data. A total of four scenes of Sentinel-2 data were used for the entire study area. Therefore, eight scenes were used in this study: four scenes captured pre or before Hurricane Irma (from December 2016 to February 2017) and four scenes captured post or after Hurricane Irma (from November 2017 to March 2018) (Table 3).

We obtained the target data by ourselves. Previous studies on mangrove mapping using semantic segmentation used manually marked labels based on visual interpretations as visually interpreted labels [32] and used machine learning methods (RF and SVM) to classify images, then used GIS tool used in postprocessing misclassification to obtain satisfactory visually interpreted labels [31]. In the present study, we combined the methods of the two aforementioned studies to obtain our visually interpreted label data by comparing the images captured before and after the hurricane. We divided the visually interpreted labels into three classes: (1) intact mangroves, comprising mangrove objects that remained intact in the images captured before and after the hurricane; (2) degraded

mangroves, comprising mangrove objects that remained intact in the images captured before the event hurricane but were degraded in the images captured after the hurricane; and (3) non-mangrove areas, comprising all objects not included in the mangrove objects (e.g., buildings, non-mangrove vegetation, bare land, and water bodies; Table 4).

**Table 3.** Sentinel-2 imagery for this study.

| Image (Scene Code) | Date Acquired (yyyy/mm/dd) |
|---|---|
| Before Hurricane Irma: | |
| Sentinel-2A (T17RMJ) | 2016/12/30 |
| Sentinel-2A (T17RNJ) | 2017/1/6 |
| Sentinel-2A (T17RMH) | 2017/2/15 |
| Sentinel-2A (T17RNH) | 2017/2/15 |
| After Hurricane Irma: | |
| Sentinel-2A (T17RMJ) | 2017/11/25 |
| Sentinel-2B (T17RNJ) | 2018/1/16 |
| Sentinel-2B (T17RMH) | 2018/1/16 |
| Sentinel-2A (T17RNH) | 2018/3/22 |

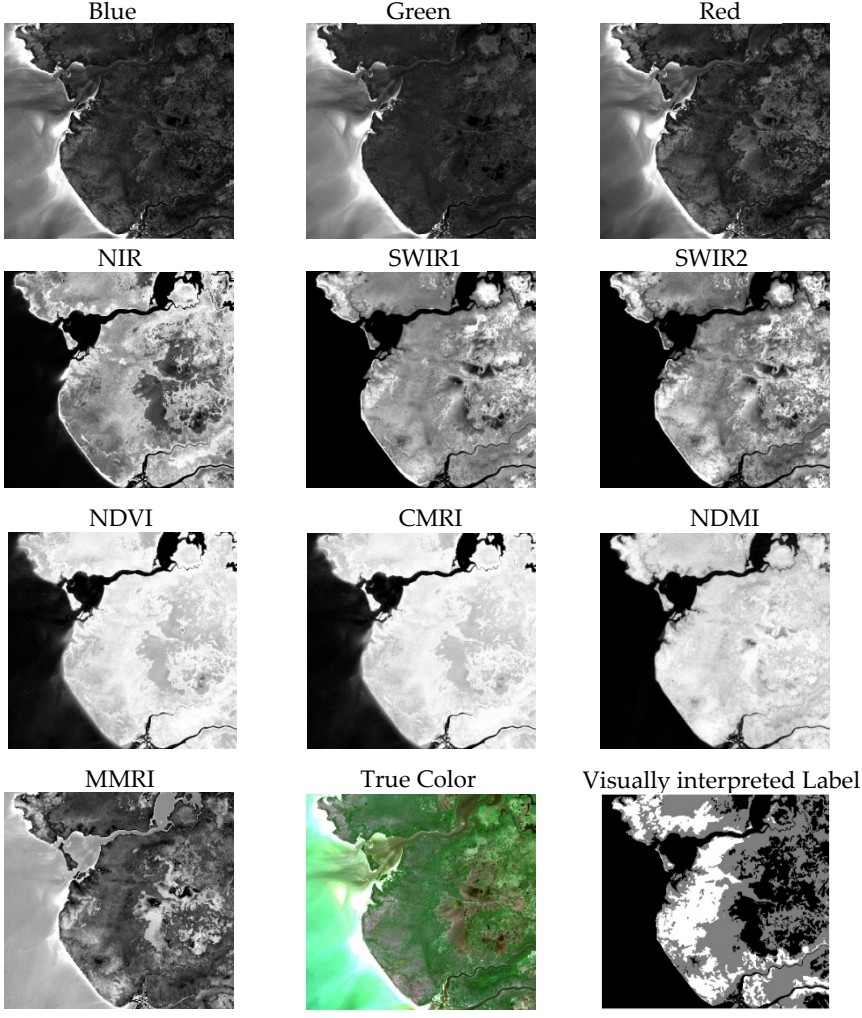

**Figure 3.** 10 input bands and visually interpreted labels in one of AOI. On the visually interpreted label, grey, white, and black color represents a mangrove, degraded mangrove, and non-mangrove class, respectively.

**Table 4.** Three classes of visually-interpreted labels.

| Classes | Image before Event | Image after Event |
| --- | --- | --- |
| Non-Mangrove | | |
| Intact Mangrove | | |
| Degraded Mangrove | | |

We divided the labeling process into four steps to obtain three classes of labeled data (Figure 4). In the first step, we manually delineated non-mangrove and mangrove objects by using the Sentinel-2 images captured before the hurricane; thus, we obtained labeled images of non-mangrove and mangrove areas. In the second step, we visually interpreted sample points and applied the RF algorithm based on the sample points to classify the non-mangrove and mangrove objects in the Sentinel-2 image captured before the hurricane. In the third step, we combined the manually delineated images of non-mangrove and mangrove areas with the classified images obtained using the RF algorithm to correct for any misclassification and obtain satisfactory images with labeled non-mangrove and mangrove areas. In the fourth step, we determined the classes of intact mangroves and degraded mangroves through manual delineation to obtain the visually interpreted labels for degraded mangrove objects based on the mangrove-labeled and Sentinel-2 images after the hurricane.

All seven AOI images including the 10 input bands and visually interpreted labels were first cropped to $64 \times 64$ sizes using a 32 pixel-step patchify library. A small patch was used to capture the spatial feature information of the mangrove objects more precisely and more easily feed the data into the model. The data set contained a total of 14,272 small patched images ($64 \times 64$), and 70% (9990 images) of the data were used for model training, 15% were used for training model validation (2141 images), and the remaining 15% were used for data testing (2141 images). We used the validation data to perform the early stopping function for the model and used the testing data to calculate the model performance metrics for accuracy assessment. The total number of pixels of each class is shown in Table 5. The non-mangrove class has a higher total number of pixels due to the non-mangrove class being a background class that has some landcover types (e.g., water bodies, vegetation non-mangrove, urban area, open area, etc.), while the intact and degraded mangrove classes have a similar total number of pixels.

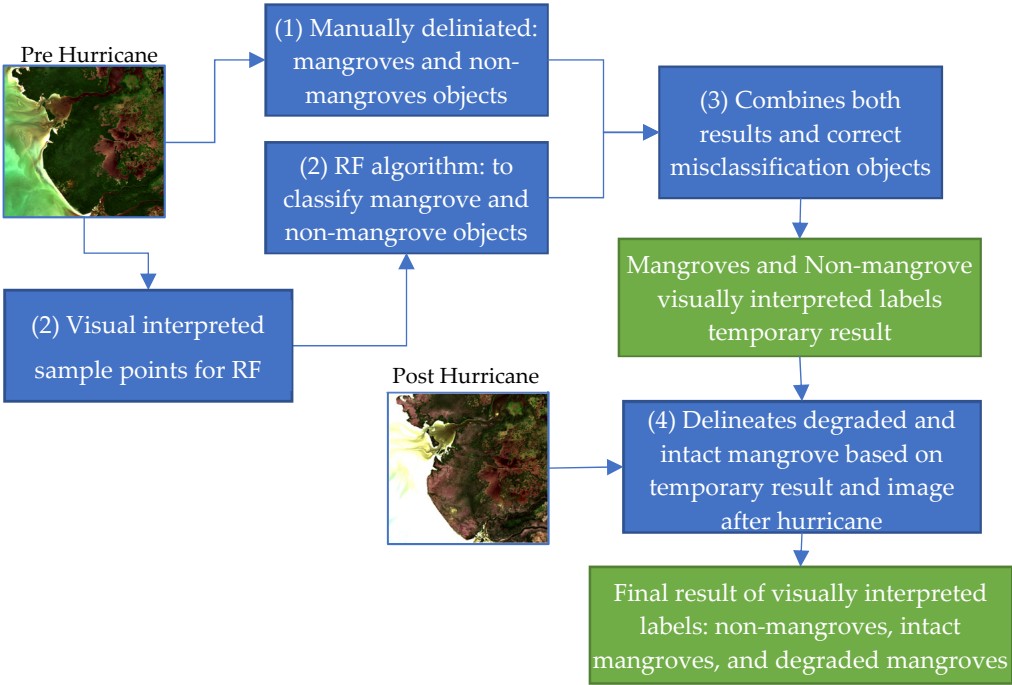

**Figure 4.** Flowchart of steps to obtain visually interpreted labels.

**Table 5.** Total number of pixels for each class in training, validation, and testing dataset.

| Class/Data | Training Data | Validation Data | Testing Data |
|---|---|---|---|
| Non-Mangrove | 35,452,388 | 7,672,797 | 7,668,533 |
| Intact Mangrove | 2,575,944 | 525,356 | 523,906 |
| Degraded Mangrove | 2,890,708 | 571,383 | 577,097 |

*2.4. Reference Samples for Map Accuracy in Whole Study Area*

The second goal of this study was to map mangrove degradation caused by Hurricane Irma along the coastal zone of southwest Florida or in the entire study area. We extrapolated the seven AOIs to the entire study area and produced a mangrove degradation map for the coastal zone of southwest Florida. We used stratified random sampling around the coastal zone to obtain reference samples for calculating the accuracy of the mangrove degradation map. Sentinel-2 and high-resolution images sourced from Google Earth before and after the hurricane were used to obtain validation point samples. A total of 1500 reference points were obtained along the coastal zone of southwest Florida: 500 reference points for the non-mangrove class, 500 reference points for the intact mangrove class, and 500 points for the degraded mangrove class (Figure 5).

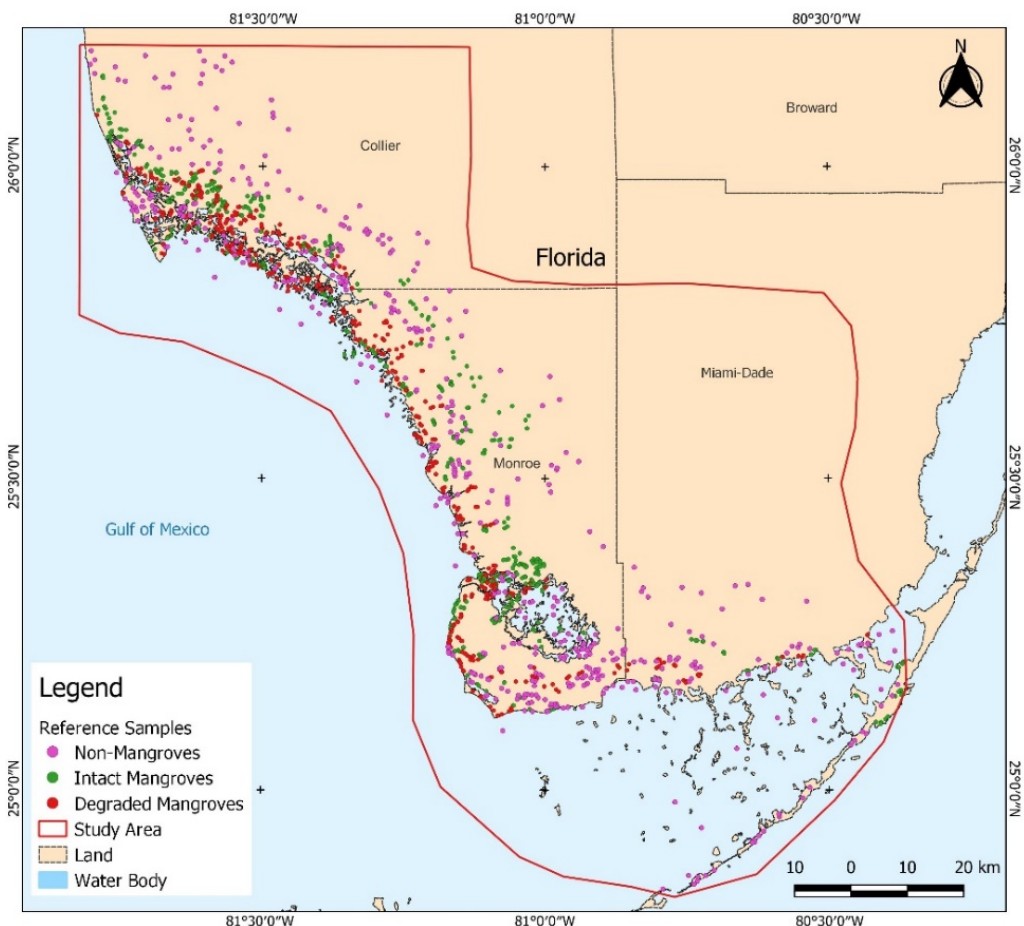

**Figure 5.** Distribution of reference samples for map accuracy in the entire study area.

### 2.5. MDPrePost-Net Architecture

This section will introduce the proposed MDPrePost-Net in detail. The proposed framework of MDPrePost-Net considers the mangrove conditions before and after Hurricane Irma simultaneously because the correlation between pre and post satellite images are salient features to determine the degraded mangrove affected by Hurricane Irma. Generally, the proposed MDPrePost-Net consists of two sub-models as shown in Figure 6, a pre-post deep feature extractor, and an FCN classifier. The pre-post deep feature extractor consists of two sub-feature extractors: the ConvLSTM feature extractor and the post feature extractor. The features extracted by the pre-post deep feature extractor are concatenated and fed into the FCN classifier model to obtain the final mangrove degradation map (non-mangrove, intact mangrove, and degraded mangrove classes). The details about sub-models in the proposed MDPrePost-Net is introduced in Sections 2.5.1 and 2.5.2.

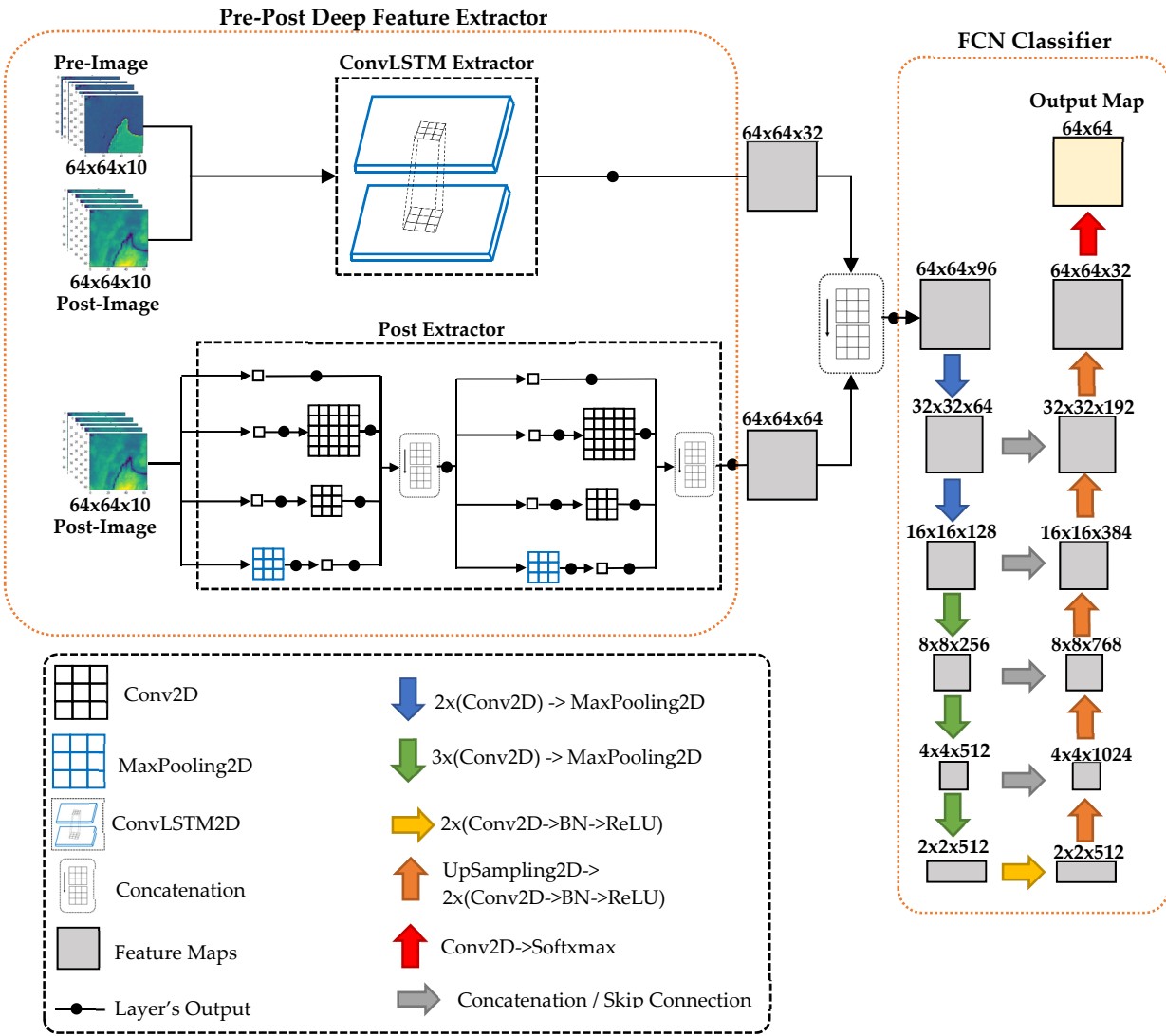

**Figure 6.** The framework of the MDPrePost-Net.

### 2.5.1. Pre-Post Deep Feature Extractor

In the proposed MDPrePost-Net, the pre-post deep feature extractor is used to extract salient features for the subsequent FCN classifier. The pre-post deep feature extractor consists of two sub-feature extractors, the first one is the ConvLSTM feature extractor used to extract spatial-spectral-temporal correlations between pre and post satellite imageries of Hurricane Irma. The other one is the post extractor used to extract spatial–spectral satellite imagery of the mangrove's condition after Hurricane Irma. Then, the features obtained by ConvLSTM and the post extractor are concatenated as spatial–spectral–temporal features that can improve the performance of the subsequent FCN classifier and mangrove mapping in the area affected by Hurricane Irma. The reason why we use two sub-feature extractors in the pre-post deep feature extractor is that, although ConvLSTM extractor can exploit the spatial–spectral–temporal correlation of pre and post satellite images, the post extractor can help to improve the spatial–spectral information in terms of the mangrove's condition after the hurricane.

The first extractor in the pre-post deep feature extractor is a spatial–spectral–temporal feature extractor based on ConvLSTM which is an extended version of the LSTM layer. The LSTM layer only considers the temporal correlation without paying attention to spatial information. To overcome this LSTM drawback, input-to-state and state-to-state transitions are performed using convolutional structures in the ConvLSTM layer. The

input of ConvLSTM is 5D tensor which can be denoted as $X_n \times T \times R \times C \times D$, where the input images are denoted as $X_n$, time or sequences as $T$, rows and columns as $R \times C$, and dimensions or channels as $D$. We use pre and post satellite imagery as the temporal or sequences image data with $T$ is 2, $R \times C$ is $64 \times 64$, and $D$ is 10 channels.

In ConvLSTM, the inner structure consists of $X_1, \ldots, X_t$ as the inputs, $C_1, \ldots, C_t$ as the cell outputs, $H_1, \ldots, H_t$ as the hidden states, and $i_t, f_t, o_t$ as the gates of the ConvLSTM are 3D tensors. Equation (1) shown the key equation of ConvLSTM with '$*$' denotes the convolutional operator and '$\circ$' denotes the Hadamard product [44].

$$
\begin{aligned}
\text{ConvLSTM} = \\
i_t &= \sigma(W_{xi}x_t + W_{hi}h_{t-1} + W_{ci} \circ c_{t-1} + b_i) \\
f_t &= \sigma\left(W_{xf}x_t + W_{hf}h_{t-1} + W_{cf} \circ c_{t-1} + b_f\right) \\
c_t &= f_t \circ c_{t-1} + i_t \circ \tanh(W_{xc}x_t + W_{hc}h_{t-1} + b_c) \\
o_t &= \sigma(W_{xo}x_t + W_{ho}h_{t-1} + W_{co} \circ c_{t-1} + b_o) \\
h_t &= o_t \circ \tanh(c_t)
\end{aligned}
\tag{1}
$$

In this ConvLSTM extractor, we use 2D kernels of size $U \times V$ is $3 \times 3$ and 32 filters map for the ConvLSTM layer. The obtained feature from ConvLSTM feature extractor is a spatial-spectral-temporal 4D tensor. Therefore, the 4D tensor obtained by the ConvLSTM feature extractor can be formulated as follows:

$$
X_{fm,\text{ConvLSTM},3,32}(T, R, C, D) = \{X_n(64, 64, 32)\}
\tag{2}
$$

where $X_{fm,\text{ConvLSTM},n,f}$ is the ConvLSTM feature map with $n$ representing the kernel size and $f$ representing the total filter.

The second feature extractor in the pre-post deep feature extractor is the spatial–spectral extractor of the mangrove's condition after Hurricane Irma, referred to as post extractor. This post extractor implements inception modules from the inception network [55] as the convolutional network to get detailed information of the mangrove's condition after the hurricane using post images data. The advantages of the inception module are that it can overcome overfitting and computational cost problems by using a $1 \times 1$ convolutional layer at the first stage before $3 \times 3$ and $5 \times 5$ convolution layers. This inception module uses three parallel $1 \times 1$ convolutional layers and one $3 \times 3$ maxpooling layer at the first stage. One $1 \times 1$ convolutional layer is directly fed into concatenation in the first stage. The second stage is the other two $1 \times 1$ convolutional layers being fed into $3 \times 3$ and $5 \times 5$ convolutional layers before concatenation, and the output of maxpooling layer fed into $1 \times 1$ convolutional layer. Each 2D convolutional in the inception module was followed by ReLU and batch normalization. The final output of the inception module is the concatenation layer that concatenates the output feature maps from the second stage and $1 \times 1$ convolutional from the first stage. We used two consecutive inception modules with dimension reductions. The input data for the post extractor are post images data with a size of $R \times C \times D_{10}$. The final output of the two inception modules with a dimension reduction can be formulated as follows:

$$
\begin{aligned}
X_{fm,Im}(R \times C \times D_{10}) &= Im_1 + Im_2 \\
Im &= S\left\{ \left[ X_{fm,1}^1, X_{fm,1}^2, X_{fm,3}^2, X_{fm,5}^2 \right] \right\}
\end{aligned}
\tag{3}
$$

where $X_{fm}$ is the 2D convolutional feature map, $Im$ is the inception module, $S$ is the concatenation layer, and $X_{fm,n}^y$ is the 2D convolutional feature map with $y$ representing the stage in the inception module and $n$ representing the kernel size of the 2D convolutional.

Generally, the aim of the pre-post deep feature extractor is to perform spatial–spectral–temporal analysis from pre and post images by using ConvLSTM. In addition, since timeliness is vital for the disaster-related research, with the aid of the post image features, more spatial–spectral information of the mangrove's condition after the hurricane are

obtained to improve the effectiveness of the pre-post deep feature extractor. The final output of the deep feature extractor is the concatenation layer that the concatenated output features from the ConvLSTM feature extractor and post feature extractor. Equation (4) shows the formula of the concatenation output feature map from this sub-model denoted as $X_{fm,sm}$.

$$X_{fm,sm} = S\left\{\left[X_{fm,\text{convLSTM}}, X_{fm,Im}\right]\right\} \tag{4}$$

### 2.5.2. FCN Classifier

The second sub-model in the proposed MDPrePost-Net is FCN architecture for the final classifier. This sub-model consists of encoder and decoder components. We adopted U-Net architecture for this sub-model. The skip connection with the concatenation layer between each block of encoder and decoder was used in this sub-model, as shown in Equation (5).

$$X_{fm,sc} = S\left\{\left[X_{fm,ec,i}, X_{fm,dc,i}\right]\right\} \tag{5}$$

where $X_{fm,sc}$ represents skip connection, $X_{fm,ec,i}$ represents each encoder block, and $X_{fm,dc,i}$ represents each decoder block.

We adopted VGG-16 architecture to construct the encoder part by removing two fully connected layers and SoftMax from VGG-16 architecture. The concatenated feature maps from the previous pre-post deep feature extractor are directly fed into the encoder part. The first and second encoder blocks consist of two 2D convolutional layers with ReLU as the activation function followed by the 2D maxpooling layer. The other three encoder blocks consist of three 2D convolutional layers with ReLU as the activation function followed by the 2D maxpooling layer. The bottom part of this sub-model after the encoder part has a feature map size of $2 \times 2$ and consists of two 2D convolutionals followed by batch normalization and ReLU. The decoder part uses the Upsampling2D layer to up-sample the feature map. Each block of the decoder part consists of Upsampling2D followed by two 2D convolutional layers followed by batch normalization and ReLU. The final 2D convolutional layer uses $1 \times 1$ kernel size and SoftMax activation function as the classifier layer. The SoftMax activation function represents a probability of each class between 0 and 1.

$$X_{fmo} = \sigma\left(X_{fm,1,1}\right)$$
$$\sigma = \frac{e^{z_i}}{\sum_{j=1}^{K} e^{z_j}} \tag{6}$$

where $X_{fmo}$ represents the final output map, $\sigma$ represents the SoftMax function, $X_{fm,1,1}$ represents the 2D convolutional with $1 \times 1$ kernel and 3 filters, $e^{z_i}$ represents the standard exponential function for the input vector, K represents the number of classes, and $e^{z_j}$ represents the standard exponential function for the output vector.

### 2.6. Algorithm Output Accuracy Assessments

We used a testing dataset (15% of the dataset) to calculate the algorithm output assessments, also known as the model evaluation. We calculated three metrics for the model evaluation: intersection over union (IoU) [56], overall accuracy (OA), and F1-score (F1) [57] for three classes (non-mangroves, intact mangroves, and degraded mangroves). IoU, also known as the Jaccard index, is widely used in semantic segmentation tasks [56]. IoU is the area of overlap between the visually interpreted labels and the predicted results divided by the area of union between the visually interpreted labels and the predicted results. Table 6 presents the formulas for the model evaluation metrics, where C, TP, TN, FP, and FN represent class, true positive, true negative, false positive, and false negative.

**Table 6.** Evaluation metrics for algorithm output assessments.

| Model Evaluation Metrics | Formula |
|---|---|
| IoU Score | $IoU = \dfrac{\lvert Ground\ truth \cap Predicted\ result \rvert}{\lvert Ground\ truth \cup Predicted\ result \rvert}$ |
| Overall Accuracy | $OA = \dfrac{\sum_{i=1}^{C} TP_i}{\sum_{i=1}^{C} (TP_i + FP_i + TN_i + FN_i)}$ |
| F1-Score | $F1 = 2 \times \dfrac{\sum_{i=1}^{C}\left(\left(\frac{TP_i}{TP_i+FP_i}\right)\times\left(\frac{TP_i}{TP_i+FN_i}\right)\right)}{\sum_{i=1}^{C}\left(\left(\frac{TP_i}{TP_i+FP_i}\right)+\left(\frac{TP_i}{TP_i+FN_i}\right)\right)}$ |

### 2.7. Map Accuracy Assessments

We used reference samples (Section 2.4) that have been collected by visual interpretation to calculate the map accuracy in the entire study area. We used OA, the Kappa score (K) [58,59], user's accuracy (UA), and producer's accuracy (PA) for the non-mangrove, intact/healthy mangrove, and degraded mangrove classes. These four metrics are commonly used to evaluate predicted mapping results against reference sample points. We used the confusion matrix to calculate those four metrics (Table 7). The OA score was calculated using the following Equation (1):

$$OA = (A + E + I)/Total\ Sample \tag{7}$$

**Table 7.** Confusion matrix for map accuracy assessments.

| | | Reference Data | | | | | |
|---|---|---|---|---|---|---|---|
| | | Non-Mangroves | Intact Mangroves | Degraded Mangroves | Rows Total | User's Accuracy | Chance Agree |
| **Predicted Data** | Non-Mangroves | A | B | C | A+B+C | A/Rows Total 1 | Rows Total 1/Total Sample |
| | Intact Mangroves | D | E | F | D+E+F | E/Rows Total 2 | Rows Total 2/Total Sample |
| | Degraded Mangroves | G | H | I | G+H+I | I/Rows Total 3 | Rows Total 3/Total Sample |
| | Columns total | A+D+G | B+E+H | C+F+I | Total Sample | | |
| | Producer's Accuracy | A/Columns Total 1 | E/Columns Total 2 | I/Columns Total 3 | OA | Kappa | |
| | Chance Agree | Columns Total 1/Total Sample | Columns Total 2/Total Sample | Columns Total 3/Total Sample | | | Total Change Agree |

In the OA formula, A, E, and I are the corrected predicted data based on reference samples. The other metric accuracy used to calculate the map accuracy is the Kappa score. To calculate the Kappa score, we needed to change the agreed value that represents the row and column total divided by the total sample for each row and column. The formula for the Kappa score was calculated in the following Equation (2):

$$Kappa = (OA - total\ change\ agree)/(1 - total\ change\ agree) \tag{8}$$

### 2.8. Implementation Details

The 10 bands (Blue, Green, Red, NIR, SWIR1, SWIR2, NDVI, CMRI, NDMI, and MMRI) of the Sentinel-2 images captured pre and post-Hurricane Irma, along with the visually interpreted labels, were first patched into $64 \times 64$ images and used as input data for our DL model. To perform the early stopping function based on a validation loss with the patience of 30 epochs, 70% of the input data were used for training and 15% were used for validation. IoU and F1 were used for validation during training, and the sum of categorical focal loss and dice loss was used to derive training loss during training. We used Adam as the optimizer with the default parameter setting and a learning rate of 0.001,

as recommended by an original Adam paper. The batch size was set to 64 for all the models. We constructed the model using Tensorflow and Keras (https://www.tensorflow.org/api_docs/python/tf/keras accessed on 1 December 2021) provided by Google in Python. The Python code was executed on a Windows 10 platform with an Intel Core i9-11900K CPU, NVIDIA GeForce RTX 3090 GPU, and 64 GB of RAM.

## 3. Results

This section presents the experimental results of the MDPrePost-Net (Section 3.1). The comparison of the MDPrePost-Net results with existing FCN architecture is drawn in Section 3.2. We investigated the effects of adding the NDVI, CMRI, NDMI, and MMRI to the input data and found that added spectral indices improved the accuracy result (Section 3.3). We extrapolated and applied the trained and proposed MDPrePost-Net to produce an intact and degraded mangrove map in the entire study area and calculate the map's accuracy. Our map shows good Kappa, producer's, user's and overall accuracy scores (Section 3.4).

### 3.1. MDPrePost-Net Results

MDPrePost-Net was trained using an early stopping schema to avoid overfitting, signifying that the training process automatically stopped if the validation loss did not improve further. The accuracy and loss curves from training and validation data from MDPrePost-Net are shown in Figure 7. The training and validation loss decreased steadily, and the training and validation IoU score and F1-Score increased steadily until the training was completed (292 total epochs). This result shows that the MDPrePost-Net completely learned the input and target data for three classes of mangrove degradation classification.

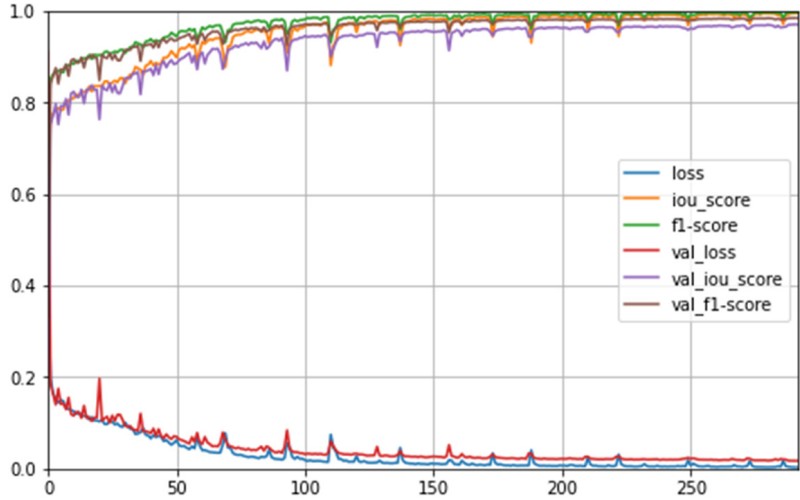

**Figure 7.** Training curve of MDPrePost-Net.

The trained model of MDPrePost-Net was evaluated by using a testing dataset that is different data from the input training data. The IoU score, F1-Score, and class accuracy for non-mangrove, intact mangrove, and degraded mangrove classes are shown in Table 8. Overall, the non-mangrove class achieved better accuracy than the intact and degraded mangrove classes. This indicates that intact and degraded mangrove classes are harder to classify than the non-mangrove class. However, the accuracy of intact and degraded mangrove classes is also high (>95%). The IoU score of intact and degraded mangrove classes is 96.47% and 96.82%, respectively. These results indicate the MDPrePost-Net successfully identifies three classes of this study with high accuracy.

**Table 8.** MDPrePost-Net IoU, F1-Score, class accuracy, and average accuracy from testing data (15% of total data).

| Accuracy Metrics | Non-Mangrove | Intact Mangrove | Degraded Mangrove |
|---|---|---|---|
| IoU | 99.82% | 96.47% | 96.82% |
| F1-Score | 99.91% | 98.21% | 98.39% |
| Class Accuracy | 99.91% | 98.31% | 98.25% |

The effects of the various total numbers of input data are tabulated in Table 9. We performed experiments by using 25%, 50%, 75%, and 100% of training data to demonstrate the effects of the total number of input data on the accuracy and IoU score, respectively. Based on the experimental results, the overall accuracy and IoU scores increase gradually in line with the increase in the total number of input data. The mean IoU decreased by 9.11% if we just used 25% of training data, while the mean IoU decreased by 5.01% and 2.32% if we just used 50% and 75% of training data, respectively. This result indicates that the total number of input data affects the accuracy scores. The fewer input data that are used, the lower accuracy scores are obtained.

**Table 9.** Effects of the total number of input data on accuracy scores, the bold values indicate the best results. NM, Mg, and DgMg stand for non-mangrove, intact mangrove, and degraded mangrove class, respectively.

| Total Input Data | Total Images (64 × 64) | Overall Acc | Mean IoU | NM IoU | Mg IoU | DgMg IoU |
|---|---|---|---|---|---|---|
| 25% of training data | 2497 | 98.53% | 88.59% | 99.27% | 82.76% | 83.73% |
| 50% of training data | 4995 | 99.10% | 92.69% | 99.56% | 88.84% | 89.68% |
| 75% of training data | 7492 | 99.44% | 95.38% | 99.73% | 92.95% | 93.46% |
| All training data | 9990 | **99.71%** | **97.70%** | **99.82%** | **96.47%** | **96.82%** |

To evaluate the effectiveness of each extractor part in the pre-post deep feature extractor, ablation studies were conducted. We removed certain parts in the first sub-model of the MDPrePost-Net to understand the contribution of each extractor part to the overall architecture based on the accuracy result. The classification results of removing the ConvLSTM extractor or the post extractor in the proposed model are shown in Table 10. The average accuracy is the averaging of Mean IoU, OA F1-Score, and overall accuracy. The results show that the ConvLSTM extractor part is the most important and effective part of the spatial–spectral–temporal deep feature extraction sub-model. If we removed the ConvLSTM extractor from the network, the average accuracy decreased by 1.13%, while the mean IoU decreased by 2.05%. This finding indicates that the spatial–spectral–temporal extracted feature from the ConvLSTM part, with pre-post data as the input, is an important feature for mangrove degradation classification affected by Hurricane Irma.

**Table 10.** Analysis of each removed part in the spatial–spectral–temporal deep feature extraction sub-model, the bold values indicate the best results. Mg and DgMg stand for intact mangrove and degraded mangrove class, respectively.

| Architecture | Mean IoU | Average Accuracy | Mg IoU | DgMg IoU |
|---|---|---|---|---|
| MDPrePost-Net | **97.70%** | **98.75%** | **96.47%** | **96.82%** |
| No Post extractor | 97.11% | 98.43% | 95.57% | 95.97% |
| No ConvLSTM extractor | 95.65% | 97.62% | 93.41% | 93.85% |

The mean IoU decreased by 0.6% if we removed the post extractor part and the IoU of intact and degraded mangrove decreased by 0.85%~0.90%. The post extractor part is less influential than the ConvLSTM part based on the accuracy metrics. However, adding the post extractor part in the network increased the average accuracy and IoU for each class. This result demonstrates the effectiveness of the two feature extractors in the proposed MDPrePost-Net.

### 3.2. Comparison with Existing Architecture

The classification result from the proposed model was compared with some well-known FCN architecture, namely FC-DenseNet [42], U-Net [40], LinkNet [41], and FPN [43]. We used the same input data and same seed random number to train the existing FCN architecture. Because the existing FCN architecture cannot handle time-series data, we stacked the pre-post image data from $(2 \times R \times C \times D_{10})$ into $(R \times C \times D_{20})$ and fed the data to the existing FCN architecture. A visual comparison of a small part from the MDPrePost result and some existing FCN architecture is presented in Figure 8. Based on the visualization analysis, the proposed model achieved improved classification compared to other existing FCN architecture. The classified map from MDPrePost-Net can clearly distinguish each class even if it has a little misclassification (non-mangrove, intact mangrove, and degraded mangrove).

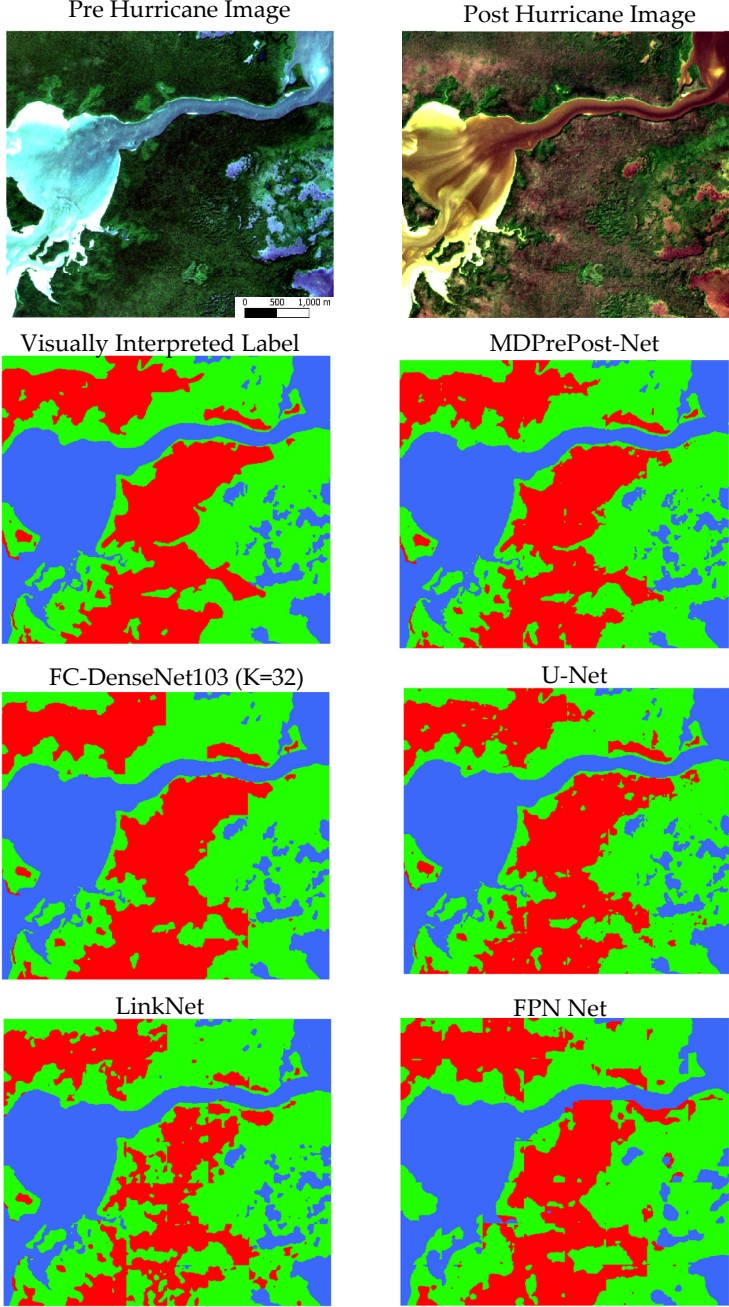

**Figure 8.** Visual comparison of the MDPrePost-Net with existing FCN architecture. K in FC-DenseNet stands for growth rate.

FC-DenseNet103 with a growth rate (K) of 32 and U-Net architecture both demonstrated good visualization results, but there exist some misclassified areas of degraded mangrove classes at the top section. On LinkNet results, the top section can better classify intact and degraded mangrove classes, but in the center until the bottom section, there exist many misclassified areas. On FPN results, the classified map result is very general and missing a lot of important details. In general, existing FCN architecture can clearly distinguish non-mangrove and mangrove objects but there exist more misclassified results in terms of intact or degraded mangroves classes when compared with MDPrePost-Net results. This result indicates the spatial–spectral–temporal extracted features from the MDPrePost-Net are very important to distinguish intact and degraded mangrove objects.

The detailed quantitative comparison including IoU score, training time, and total parameters as shown in Table 11. The proposed MDPrePost-Net outperforms other existing FCN architecture in terms of mean IoU and IoU scores for each class. The total parameter of the proposed model is larger than LinkNet and FPN-Net, and the training time is also longer than LinkNet and FPN-Net but is still acceptable due to the higher accuracy and reasonable total training time that is lower than one hour. LinkNet has a very fast training time and fewer total parameters due to LinkNet aiming for real-time semantic segmentation.

**Table 11.** Quantitative comparison of the MDPrePost-Net with existing FCN architecture, the bold values indicate the proposed model results. K in FC-DenseNet stands for growth rate.

| Architecture | Mean IoU | IoU Non-Mg | IoU Mg | IoU DgMg | Time Per-Epoch (s) | Training Times (hh:mm) | Total Parameters (millions) |
|---|---|---|---|---|---|---|---|
| MDPrePost-Net | **97.70%** | **99.82%** | **96.47%** | **96.82%** | 16 | 01:17 | 23.8 |
| FC-DenseNet103 (K = 32) | 93.00% | 99.63% | 89.24% | 90.13% | 59 | 03:22 | 35.2 |
| U-Net | 92.58% | 99.62% | 88.57% | 89.54% | 13 | 01:14 | 31.4 |
| LinkNet | 92.19% | 99.42% | 88.24% | 88.93% | 5 | 0:15 | 11.6 |
| FPN-Net | 82.27% | 98.46% | 73.08% | 75.28% | 10 | 0:29 | 17.6 |

In general, almost all the existing architecture achieved a high mean IoU of more than 90%. Only FPN achieved a mean IoU lower than 90%. Based on the IoU score of the non-mangrove class, all existing FCN achieved a very high IoU score (>98%), which indicates that all the existing FCN can clearly distinguish mangrove and non-mangrove objects based on the visual analysis. FC-DenseNet103 (K = 32) achieved the second-best place in terms of mean IoU but the IoU of the intact mangrove was still lower than 90%, and the total training time and total parameter of FC-DenseNet103 were the longest and biggest. In the U-Net result, the training time is almost the same as the proposed MDPrePost-Net, but U-Net has IoU intact and degraded mangrove scores of lower than 90%. Based on this result, MDPrePost-Net successfully outperforms some existing FCN architecture in terms of mangrove degradation affected by Hurricane Irma using temporal satellite data.

### 3.3. Effects of Vegetation and Mangrove Indices on the Results

This section demonstrates the effect of the different total number of input bands in the MDPrePost-Net. In the first experiment, we used only true-color images or the blue, green, and red bands of Sentinel-2 as the input data. A true-color image is widely used for DL-based image classification. In the second experiment, we used true-color images with the NIR band (blue, green, red, NIR). In the third experiment, we used true-color images with the NIR and SWIR bands (blue, green, red, NIR, SWIR1, and SWIR2). The third experiment was conducted to study the effects of including the SWIR bands in the classification of intact and degraded mangroves because the SWIR band is sensitive to wet objects. The last experiment was conducted using the original input data to prove that the 10 input bands (including the NDVI, CMRI, NDMI, and MMRI) could improve the accuracy of the classification of mangroves and degraded mangroves.

Table 12 presents the accuracy assessment results (i.e., mean IoU, overall F1, OA, and average accuracy) for the three classes (non-mangroves, mangroves, and degraded mangroves) as well as the IoU result for each class. For the first experiment using RGB data, the mean IoU was 95.54%, the IoU value for the mangrove class was 93.13%, and the IoU value for the degraded mangrove class was 93.76%. This finding shows that the proposed MDPrePost-Net model has satisfactory accuracy for the true-color Sentinel-2 images. For the second and third experiments, the addition of the NIR, SWIR1, and SWIR2 bands improved the accuracy value. Specifically, the average accuracy increased by approximately 0.57%, the IoU value for the intact mangrove class increased by 1.69%, and the IoU value for the degraded mangrove class increased by approximately 1.37%. For the final experiment involving the 10 input bands, the results revealed that the added vegetation and mangrove indices considerably improved the classification accuracy.

**Table 12.** Accuracy results of added vegetation and mangrove indices, the bold values indicate the best results.

| Input Bands | Mean IoU | OA F1-Score | OA Accuracy | Average Accuracy | IoU Non-Mg | IoU Mg | IoU Dg Mg |
|---|---|---|---|---|---|---|---|
| RGB | 95.54% | 97.69% | 99.45% | 97.56% | 99.71% | 93.13% | 93.76% |
| R,G,B,NIR | 96.25% | 98.07% | 99.55% | 97.96% | 99.79% | 94.25% | 94.70% |
| R,G,B,NIR,SWIR1,SWIR2 | 96.58% | 98.25% | 99.58% | 98.13% | 99.78% | 94.82% | 95.13% |
| All 10 input bands | **97.70%** | **98.83%** | **99.71%** | **98.75%** | **99.82%** | **96.47%** | **96.82%** |

### 3.4. Mangrove Degradation Map in Southwest Florida

The second goal of this study is to map intact/healthy mangrove and degraded mangrove forests along the southwest Florida coastal zone effectively and efficiently because southwest Florida has a long and wide coastal zone. The most effective, efficient, and accurate method is needed to know the impact of Hurricane Irma on the mangrove forests area. The trained proposed MDPrePost-Net was used to produce the non-mangrove, intact/healthy, and degraded mangrove map along the southwest Florida coastal zone.

We used a confusion matrix to calculate the map accuracy based on reference samples: 1500 reference point samples, 500 samples of non-mangrove, 500 samples of intact/healthy mangroves, and 500 samples of degraded mangrove classes (Table 13). The overall accuracy, user's accuracy, producer's accuracy, and kappa score have been calculated for map accuracy based on the confusion matrix. The result shows that the overall accuracy is 0.9733. The user's accuracy and producer's accuracy of non-mangrove classes are 0.998 and 0.978, intact mangrove classes are 0.9680 and 0.9622, while degraded mangrove classes are 0.9540 and 0.9794, respectively. Based on this result, our mangrove degradation map affected by Hurricane Irma has a very good result with a kappa score of 0.960. Based on the calculation of the map accuracy, the MDPrePost-Net achieved a very good result in terms of mapping the intact and degraded mangroves affected by Hurricane Irma in the large study area.

The result of the non-mangrove, intact and degraded mangrove map along the southwest Florida coastal zone is shown in Figure 9. Based on the map, almost the entire mangrove forest which is directly adjacent to the shoreline was degraded. The degraded mangrove areas include several degraded mangroves or slightly degraded mangroves. The purple box on the map shows zoom areas that indicate severely degraded mangrove areas. Based on the zoom areas, we can see that our model can clearly distinguish intact/healthy mangrove, degraded mangrove, and non-mangrove areas. The map shows in the lower north and northwest study area that there are many degraded mangrove areas. Based on the map, we can see the pattern of intact/healthy mangrove and degraded mangrove areas. Most of the intact mangrove areas are behind the degraded mangrove areas, which indicates that the mangrove area in front of it can protect against Hurricane Irma. We can see the mangrove areas bordering the land are not degraded. These results show that mangroves can protect the coastline area from hurricanes.

**Table 13.** Confusion matrix of map accuracy with overall accuracy, kappa score, user's accuracy and producer's accuracy from non-mangrove, intact mangrove, and degraded mangrove classes in the whole study area.

|  |  | Reference Data | | | | |
|---|---|---|---|---|---|---|
|  |  | Non-Mangroves | Intact Mangrove | Degraded Mangrove | Rows Total | User's Accuracy |
| Predicted Data | Non-Mangrove | 499 | 0 | 1 | 500 | 0.9980 |
|  | Intact Mangrove | 7 | 484 | 9 | 500 | 0.9680 |
|  | Degraded Mangrove | 4 | 19 | 477 | 500 | 0.9540 |
|  | Column total | 510 | 503 | 487 | 1500 | |
|  | Producer's Accuracy | 0.9784 | 0.9622 | 0.9794 | Overall Acc = 0.9733 | |
|  |  |  |  |  | Kappa = 0.960 | |

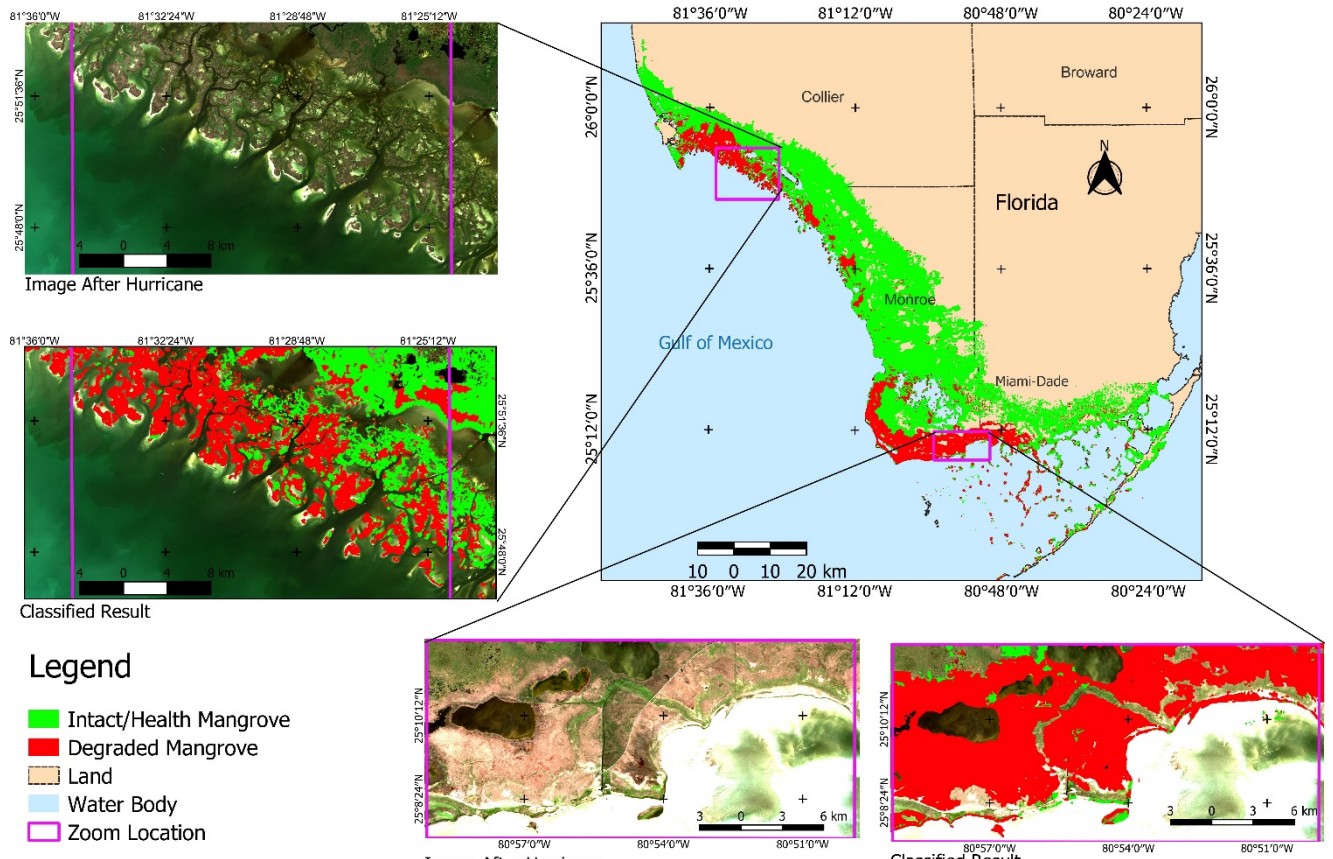

**Figure 9.** Intact and degraded mangrove map affected by Hurricane Irma in southwest Florida 2017 using the proposed model.

## 4. Discussion

Ecosystem degradation is a principal factor behind biodiversity loss [14]. Mapping of mangrove degradation is very important to know the status of the mangroves because mangroves have many benefits [2–5]. In September 2017, southwest Florida was hit by Hurricane Irma (a category 3 hurricane) which degraded mangrove forests [20]. By developing advanced technology in satellite and computer science, remote sensing imagery have been widely used for mangrove mapping using object-based classification [23], RF algorithm [24,26], CART algorithm [25], Capsules-U-Net [31], Me-Net [32], and convolutional neural network [33]. Some previous studies used satellite imagery for mangrove degradation mapping in southwest Florida affected by Hurricane Irma by using machine learning algorithms [20–22]. This study proposed the deep learning model that can be used for mangrove degradation mapping using Sentinel-2 data by considering the spatial–spectral–

temporal relationship between images before hurricanes and images after hurricanes and producing a map of intact and degraded mangrove areas affected by Hurricane Irma along southwest Florida.

In this study, we consider two kinds of satellite imagery for mangrove degradation mapping. The first one is pre-image referring to images before Hurricane Irma and the second one is post-image referring to images after Hurricane Irma. We assume the degraded mangrove class is the mangrove object in the image before Hurricane Irma which is then degraded in the image after Hurricane Irma because the degraded mangrove area in this study was affected by Hurricane Irma. The proposed MDPrePost-Net consists of two sub-models, a pre-post deep feature extractor and an FCN classifier. The MDPrePost-Net considers the relationship between the pre and post-hurricane events by using the ConvLSTM feature extractor in the deep feature extraction sub-model. Previous studies used the combination of ConvLSTM with semantic segmentation for precipitation nowcasting [44], extracting spatiotemporal relationships and classifying SLAR images [45] and extracting spatiotemporal relationships of video sequences classifications [46,47]. In this study, we used the ConvLSTM part for extracting the spatiotemporal relationship between the images captured before the hurricane and those captured after the hurricane. The post extractor also considered the spatial–spectral information of images after the hurricane that was more correlated with the degraded mangrove area that was affected by Hurricane Irma.

Based on the classification result, our proposed model achieved good results in terms of accuracy metrics and acceptable training time. Our proposed model has a mean IoU score, F1-Score, and overall accuracy of 97.70%, 98.83%, and 99.71%, respectively. The total training time of our proposed model is about one hour (01:17). We compared our MDPrePost-Net results with other existing FCN architecture (FC-DenseNet [42], U-Net [40], LinkNet [41], and FPN [43]). We used the same input data to train all the existing FCN architecture. All the existing architecture can clearly distinguish non-mangrove objects with IoU scores of more than 90% for the existing FCN architectures. However, the existing FCN architectures have more misclassified results in terms of intact and degraded mangrove classes, and the average IoU scores for both classes are lower than 90%. These results were caused by the existing FCN architecture not being specially designed to acquire a spatial–spectral–temporal relationship of pre-post images. Our proposed model considered the spatial–spectral–temporal relationship and increased the IoU scores of intact and degraded mangrove objects.

A previous study revealed that the difference in input bands affects the accuracy result of mangrove classification [32]. We considered the effects of vegetation and mangrove indices in the accuracy results because mangroves are a unique form of vegetation with unique spectral characteristics, such as wet vegetation features. The SWIR band is useful for distinguishing wet objects. Some mangrove indices are designed according to spectral mangrove characteristics. Sentinel-2 can produce vegetation and mangrove indices. The CMRI, NDMI, and MMRI make up the vegetation index which has a good capability to distinguish mangrove objects [24,51,52]. In our original input data for this study, we used 10 input bands (blue, green, red, NIR, SWIR1, SWIR2, NDVI, CMRI, NDMI, and MMRI) with vegetation and mangrove indices. Our results in Section 3.3 reveal the mean IoU of 10 input bands increased by approximately 2.16% relative to that observed for the first experiment (R, G, B). This significant improvement was observed in the IoU value for the intact and degraded mangrove classes. The IoU value for the intact mangrove class increased by nearly 3.34%, and for the degraded mangrove class, it increased by approximately 3.06%. Based on this result, the addition of vegetation and mangrove indices increased the accuracy result.

We used the proposed MDPrePost-Net with 10 input bands to make an intact and degraded mangroves map affected by Hurricane Irma along the southwest Florida coastal zone. The confusion matrix was used to calculate the map accuracy based on reference samples. The kappa score from our proposed model result is 0.960 and is included in the almost perfect agreement category based on Landis and Koch categorizations [58].

Previous studies of mangrove degradation affected by Hurricane Irma in some areas of southwest Florida have been performed by using WorldView-2 and Landsat data which are different data from our input data in this research. The ecological conceptual model with annual metrics before and after Hurricane Irma has been used to calculate the annual NDVI mean, NDVI standard deviation, Normalize Difference Moisture Index (NorDMI) mean, NorDMI standard deviation and NDWI mean from Landsat data, and the RF algorithm for intact and degraded mangrove classification [22]. The time series WorldView-2 data (2010, 2016, 2017, and 2018) with the decision tree (DT) algorithm has been used to classify damaged mangrove forests [20], while WorldView-2 data (2018) has been used for mapping damage inflicted by Hurricane Irma using SVM, DT, and neural network [21]. Our proposed method considered the spatial–spectral–temporal relationship of pre-post Sentinel-2 imagery by using a pre-post deep feature extractor.

The comparison of the map accuracy from our proposed model with other existing results is shown in Table 14. Lee et al. conducted mangrove degradation research in the small area of Shark River in southwest Florida by using the RF algorithm and Landsat image data. They achieved an OA of 79.1% [22]. McCarthy et al. applied some machine learning algorithms (DT, NN, and SVM) to mangrove degradation research in Rookery Bay NEER in southwest Florida using WorldView-2 imagery and achieved an OA around 82%–85% [20,21]. The non-mangrove class from DT, NN, and SVM results in Table 12 have come from averaging soil, upland, and water classes. Our MDPrePost-Net results revealed the pre-post deep feature extractor significantly improved the map accuracy of intact mangrove and degraded mangrove classes.

**Table 14.** Map accuracy comparison of our proposed model with other models. N-Mg, Int-Mg, Dg-Mg stand for non-mangrove, intact mangrove, and degraded mangrove class, respectively.

| Model | UA and PA of N-Mg (%) | UA and PA of Int-Mg (%) | UA and PA of Dg-Mg (%) | OA (%) | Image Data | References |
|---|---|---|---|---|---|---|
| Our proposed model | **99.8 and 97.8** | **96.8 and 96.2** | **95.4 and 97.9** | **97.3** | Sentinel-2 | - |
| Neural Network | 89.3 and 89 | 80 and 88 | 75 and 57 | 85 | WorldView-2 | [21] |
| Decision Tree | 89.7 and 86.7 | 73 and 91 | 62 and 38 | 83 | WorldView-2 | [21] |
| Decision Tree | 87.7 and 84.7 | 78 and 77 | 56 and 54 | 82 | WorldView-2 | [20] |
| Support Vector Machine | 88.7 and 87 | 77 and 86 | 59 and 62 | 83 | WorldView-2 | [21] |
| Random Forest | - | 85.7 and 72.3 | 73.9 and 86.7 | 79.1 | Landsat | [22] |

A total of 22% of mangrove areas were lost from spring 2016 to fall 2018 in Rookery Bay NEER and most were affected by Hurricane Irma [20], while 97.4% of mangrove areas have been degraded by Hurricane Irma in the small area of the Shark River case-study region [22]. These two areas are part of our study area. Our study reported that the total mangrove area, including intact/healthy and degraded mangroves in our study area, which is 153,933.36 Ha, has losses of 6.96% from Global Mangrove Forest reported by Giri, et al in 2000 [1]. Based on our result, 26.64% (41,008.66 Ha) of the mangrove area has been degraded by Hurricane Irma, and the other 73.36% (112,924.70 Ha) of the mangrove area is still intact. Our result indicated that extreme climates event [10] and hurricanes [13] can degrade mangrove forests.

Our results of total intact and degraded mangrove areas are affected by the presence of cloud cover in some Sentinel-2 data that we used. The acquisition time of Sentinel-2 used in this research also affects the total or degraded mangrove areas because degraded mangrove areas can recover in a certain period of time. McCarthy et al. reported a total of 1.73 km$^2$ of mangroves recovered in their study area 6 months after Hurricane Irma [20]. The last data from Sentinel-2 we used in this research were in March 2018 which is 6 months after the Hurricane Irma event. The selection of this data is based on the percentage of cloud cover to minimize discrepancies in results. The recovery time of degraded mangroves affected by hurricanes needs more attention for future research.

## 5. Conclusions

In September 2017, Hurricane Irma hit the coastal zone of southwest Florida and caused mangrove degradation. This study aimed to propose an end-to-end deep learning network that exploited the spatial–spectral–temporal relationship between images before the hurricane and images after the hurricane, referred to as the MDPrePost-Net. Generally, our proposed model consists of two sub-models, a pre-post feature extraction and the FCN classifier. We used pre and post Sentinel-2 to extract the spatial–spectral–temporal relationship of intact and degraded mangroves that were affected by Hurricane Irma by using the ConvLSTM feature extractor and used post image Sentinel-2 data to extract spatial–spectral of the mangrove condition after the hurricane by using a post feature extractor. Then the extracted spatial–spectral–temporal features were fed into an FCN classifier. A total of 10 input bands were used in this study (blue, green, red, NIR, SWIR-1, SWIR-2, NDVI, CMRI, NDMI, and MMRI bands). Our results reveal that the adding of vegetation and mangrove indices improves the accuracy of intact and degraded mangrove classes.

The experimental results demonstrate that the proposed model achieved better results in terms of accuracy metrics than the existing FCN architecture (FC-DenseNet, U-Net, Link-Net, and FPN). The proposed model also has an acceptable training time (01:17) or 16 seconds of each epoch. In addition, the proposed model achieved the algorithm output mean IoU, F1-Score, and OA of 97.70%, 98.83%, and 99.71%, respectively. We used the proposed model to produce an intact/healthy and degraded mangrove map affected by Hurricane Irma in 2017 along the southwest Florida coastal zone. The overall map accuracy of non-mangrove, intact/healthy and degraded mangrove classes is 0.9733 with a kappa score of 0.960. This study reported that 26.64% (41,008.66 Ha) of mangrove areas have been degraded by Hurricane Irma in September 2017, and the other 73.36% (112,924.70 Ha) of mangrove areas are still intact.

**Author Contributions:** Conceptualization, I.J.; Data curation, C.-H.C.; Formal analysis, T.T.; Investigation, T.T. and Y.-N.C.; Methodology, I.J., T.T. and Y.-N.C.; Project administration, Y.-N.C.; Resources, I.J.; Software, C.-H.C. and C.-L.H.; Supervision, Y.-N.C.; Validation, T.T. and C.-L.H.; Writing—original draft, I.J.; Writing—review & editing, Y.-N.C., C.-H.C. and C.-L.H. All authors have read and agreed to the published version of the manuscript.

**Funding:** The work was funded by supported by the Ministry of Science and Technology under grant no. MOST 110-2625-M-008-005.

**Data Availability Statement:** The Sentinel-2 images are available at the URL: https://scihub.copernicus.eu/dhus/ (accessed on 1 December 2021). While visually interpreted labels and derived mangrove degradation map are available by the request.

**Acknowledgments:** The authors would like to thank the European Space Agency (ESA) for freely providing Sentinel-2 data. We also thank the TensorFlow team (https://www.tensorflow.org/ accessed on 1 December 2021) and segmentation_model team (https://github.com/qubvel/segmentation_models accessed on 1 December 2021) that provided deep learning package.

**Conflicts of Interest:** The authors declare no conflict of interest.

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
