# Peer review of "MDPrePost-Net: A Spatial-Spectral-Temporal Fully Convolutional Network for Mapping of Mangrove Degradation Affected by Hurricane Irma 2017 Using Sentinel-2 Data"

_remotesensing, doi:10.3390/rs13245042_

Round 1

Reviewer 1 Report

The paper is well organized and presents results understandable thanks to the rich background provided by the numerous references.

As a general comment, I would suggest the authors to address some language mistakes (mostly typos but also grammar errors, i.e. lines 124-127 need to be rephrased - the structure of the sentence is quite unclear, typo in Figure 1 caption - porcessing instead of processing, etc.)

In addition, I have two more comments/questions:

1) In Sec 2.5.2 line 403 you explain that the final layer of the FCN classifier is a 2D convolutional layer with 3 filters (due to 3 classes) and 3x3 kernel. Looking at the structure of the network, I see that the padding option was used in order to obtain the final output of the same size "xy" of the previous layer. Have you tried to use 1x1 at the last stage instead of 3x3 kernel? Especially on the borders, you might not want to use convolution on "0-padded" pixels to improve the efficiency in these particular parts of the images. Is there any specific reason why you used 3x3 kernel at the last stage instead?

2) I couldn't find any reference to the dataset's classes population. Since this study is pixel-based, it could be useful to add this information (is the dataset class-balanced?). Indeed, the definition and usage of Overall Accuracy score is tricky in some points, as it seems to refer to actual accuracy score (from table 5) but in Equation (7) it seems to use class-balanced intermediate scores (A E and I).

I think that the paper is overall suitable for publication, once the English review has been operated along with the two additional comments.

Author Response

First, we like to express our grateful acknowledgement for the helpful comments of the Editor and referees in improving the earlier version of this manuscript. Our response is as the attached file, thank you.

Reviewer 2 Report

Generally, this paper is well-organized, and the results are convincing. My comments are as follows:

  1. Why do you not conduct this work after the Hurricane Irma in 2017? Timeliness is vital for the disaster-related research.
  2. You employed the LSTM architecture in your model, so strictly it is not a fully convolutional network. It is advised to remove the “fully convolutional” in the title to reduce the confusion.
  3. The author motioned in the Introduction part that “the greatest challenge of mangrove mapping is to deal with the marred condition of mangrove forests and their wide distribution within a location”. And lots of machine learning methods are summarized. My question is how these methods dealt with this challenge? An explicit explanation should be supplemented. In other words, what did other researches do to deal with this challenge in the mangrove mapping?
  4. In the data preparing part, the author used the RF as an auxiliary tool in the second step to get the mangrove and none-mangrove objects. The RF is a supervised method, so is the manually delineated labels served as the ground truth for the RF training? Did you test the effect if you only use the manually delineated labels instead of the RF classification?
  5. There are too many unnecessary basics of the LST and CNN presented in the model architecture part, please consider to simplify the introduction of basic deep learning models to highlight your own model architecture.
  6. The author chooses four images before and after the hurricane respectively in the experiment. How does the number of input images effect the classification accuracy? If you reduce or add some images, would it lower or improve the accuracy?

Author Response

(The authors gave the same response as above.)

Round 2

Reviewer 2 Report

This paper is well-presented after revision and can be accepted for publication.